# Orchestrated Action of AMPK Activation and Combined VEGF/PD-1 Blockade with Lipid Metabolic Tunning as Multi-Target Therapeutics against Ovarian Cancers

**DOI:** 10.3390/ijms23126857

**Published:** 2022-06-20

**Authors:** Mingo M. H. Yung, Michelle K. Y. Siu, Hextan Y. S. Ngan, David W. Chan, Karen K. L. Chan

**Affiliations:** 1Department of Obstetrics & Gynaecology, Li Ka Shing Faculty of Medicine, The University of Hong Kong, Hong Kong SAR, China; h1094157@connect.hku.hk (M.M.H.Y.); mkysiu@gmail.com (M.K.Y.S.); hysngan@hku.hk (H.Y.S.N.); 2School of Biomedical Sciences, The Chinese University of Hong Kong, Hong Kong SAR, China

**Keywords:** AMPK, ferroptosis, VEGF, PD-1 blockade, polyunsaturated fatty acids, tumor microenvironment, cancer metabolism, ovarian cancer

## Abstract

Ovarian cancer is one of the most lethal gynecological malignancies worldwide, and chemoresistance is a critical obstacle in the clinical management of the disease. Recent studies have suggested that exploiting cancer cell metabolism by applying AMP-activated protein kinase (AMPK)-activating agents and distinctive adjuvant targeted therapies can be a plausible alternative approach in cancer treatment. Therefore, the perspectives about the combination of AMPK activators together with VEGF/PD-1 blockade as a dual-targeted therapy against ovarian cancer were discussed herein. Additionally, ferroptosis, a non-apoptotic regulated cell death triggered by the availability of redox-active iron, have been proposed to be governed by multiple layers of metabolic signalings and can be synergized with immunotherapies. To this end, ferroptosis initiating therapies (FITs) and metabolic rewiring and immunotherapeutic approaches may have substantial clinical potential in combating ovarian cancer development and progression. It is hoped that the viewpoints deliberated in this review would accelerate the translation of remedial concepts into clinical trials and improve the effectiveness of ovarian cancer treatment.

## 1. Introduction—A Rapid Glance at Ovarian Cancer

Ovarian cancer is one of the most common and lethal gynecological malignancies [1,2]. Most patients (~65%) are poorly diagnosed until the late stages when cancer has already metastasized beyond the confines of the ovary [1,3,4]. Advanced ovarian cancer, especially from ovarian clear cell carcinoma (OCCC) subtype, is characterized by higher chemoresistance and aggressiveness [5]. Although platinum-taxane-based chemotherapy is the foremost anti-cancer curative for ovarian cancer, chemoresistance is still a critical obstacle to successful clinical management [6,7]. The emergence of drug resistance, especially in micrometastases, has rendered a broad range of the currently platinum-based chemotherapy regimens (e.g., Cisplatin, Carboplatin, Paclitaxel, etc.) ineffective. The majority of patients acquire resistance to conventional chemotherapeutics and experience aggressive tumor recurrence at a median of 15 months from diagnosis [8,9,10,11], and the five-year survival rate remains less than 40% [12]. The prevalent phenomenon is mainly attributed to the intrinsic dysregulations of the ovarian cancer cells, in which tumor evolution is involved through genetic/epigenetic alterations cooperated by the modification of key signaling pathways to adapt to the tumor microenvironment (TME) during tumor development and progression [13,14]. Therefore, a better understanding of ovarian tumor biology and exploring novel druggable targets for combating the minimal residual disease (MRD) is highly prioritized.

On the other hand, immunotherapy, which aroused compelling interests in clinical oncology during recent years, has potentially experienced encouraging outcomes in various malignancies such as melanoma, non-small cell lung cancer (NSCLC), kidney, and kidney and urothelial cancers [15,16,17]. Nevertheless, ovarian cancer has been regarded as a “cold tumor” of immunological treatments [15]. For instance, the treatments of utilizing monoclonal antibodies to neutralize single-antigens like CTLA-4, PD-1, and PD-L1 exhibited modest results in ovarian cancer patients with a median response rate of around 15% [15,18,19,20,21]. The immune-excluded characteristic of ovarian tumors is usually driven by the modification of TME and the presence of inhibitory cells such as myeloid-derived suppressor cells (MDSCs) that prevent effector CD8^+^ T cells from infiltrating the tumor islets, even if they exist in the stroma [15,22]. There is a growing number of reports indicating that lipid-enriched malignant ascites not only act as the distinct TME supplying cytokines, chemokines, microvesicles, small molecular metabolites, and bioactive lipids etc. for aggressive tumor growth and peritoneal dissemination of ovarian cancer [23,24], but also impose metabolic barriers such as hypoxia and immunosuppressive metabolites especially unsaturated fatty acids (UFAs) on antitumor immunity [25,26]. Therefore, variation in the induction of immune response contributes to the uncertainty of the therapeutic quality. In addition, a minimal perturbation of conformational epitopes on native proteins will cause a failure in recognizing the corresponding antigen by a monoclonal antibody [27]. It is worth noting that further investigations are necessary to elucidate the molecular mechanisms underlying the interactions between immune response and metabolic rewiring within the TME of ovarian cancer. In addition, more attention should be paid to novel strategies with a special focus on the combination of alternative therapeutic agents and immunotherapeutics that may help to potentiate the positive effects of the current anti-cancer remedy.

## 2. Targeting AMPK as an Alternative Approach to Combat Ovarian Cancer

Metabolic reprogramming is one of the hallmarks of cancer development and progression, and hence cancer cells have to expand nutritional requirements to sustain intensive growth [28]. Emerging evidence has confirmed that targeting cancer cell metabolism is a promising therapeutic approach in human cancers [29]. AMP-activated protein kinase (AMPK), situated in the capacity of the metabolic checkpoint, plays a crucial role in regulating this molecular adaptation. Phosphorylation of AMPK in response to pharmaceutical AMPK activators has been shown to exhibit remarkable antitumor effects on different types of human carcinoma cells [30,31,32]. Indeed, our previous investigations demonstrated that AMPK activators could repress cervical cancer cell growth harboring with/without liver kinase B1 (LKB1) [33] by reducing AKT/FOXO3a/FOXM1 signaling [33,34] as well as DVL3/WNT/β-catenin signaling [35]. More importantly, we have recently reported that bitter melon extract (BME) and its bioactive protein Momordica anti-HIV protein (MAP30) function as a natural AMPK activator inducing AMPK activity through CaMKKβ signaling in an AMP-independent manner, which in turn inhibited both mTOR/p70S6K and AKT/ERK/FOXM1 signals in ovarian cancer cells (Figure 1). These findings suggest that targeting AMPK represents a possible therapeutic approach in advanced ovarian cancers [12,36]. However, solid tumors are heterogeneous and structurally complex [37,38]. Not only the cancer cells per se are in heterogeneous cell populations with different functional properties, but also the tumor tissues comprise malignant cells and tumor-associated stromal cells (TASCs) (i.e., fibroblasts and inflammatory cells) which secrete many pro-tumorigenic factors to recruit additional tumor and pro-tumorigenic cells to the TME [39,40]. Therefore, it is still believed that other mechanisms are being exploited in tumor development and progression. The exploration of these possible mechanisms will assist in developing better interventions while further revitalizing the efficacy of existing regimes.

## 3. Limitations of Using AMPK Activators in Treating Malignancies

AMPK is a known cellular metabolic sensor that plays a significant role in controlling energy homeostasis in response to external stresses. There is already a growing interest in the therapeutic exploitation of the AMPK pathway for cancer therapy. However, recent propositions proposed that malignant cells might also utilize AMPK activation as a survival strategy to confront energy or microenvironmental stresses. These hypotheses underscore the complexity of the cellular function of AMPK in maintaining energy homeostasis under different pathological conditions. The metabolic tumor-suppressive function of AMPK might be overridden by stress or oncogenic signals in cancer cells. While numerous findings have reported that the overexpression of AMPK inhibits tumor growth and metastasis, a growing body of evidence has indeed proposed that activation of AMPK in cancer cells might support their survival, particularly at the initial stages of tumor development [29,41,42,43]. When a cancer cell cluster expands its mass at the early stages without adequate formation of blood vessels, the hypoxic microenvironment likely stimulates the expression of VEGF, which induces angiogenesis to replenish blood and nutrient supplies. How AMPK is activated and exerts a protective effect at this step by promoting autophagy has been recently described. On the other hand, a computational model has recently been established to stimulate the effects of pharmacological maneuvers that target key metabolic signaling nodes, with a specific focus on AMPK, indicating that AMPK inactivation was expected to have the beneficial effect of limiting cancer cells growth upon chronic nutrient deprivation [44]. Serval studies also consistently reported a more susceptible cell death induced by glucose starvation or extracellular matrix detachment in human cancer cells upon suppression of AMPK or depletion of its upstream kinase LKB1, inferring that AMPK activation might help protect against these insults [45,46]. Evidence that AMPK promoted tumorigenesis was likewise observed in in vivo model of NSCLC in which AMPKα1α2 double knockout together with mutant K-RAS and P53 deletion was found to retard the magnitude and the number of tumors [47]. However, it can be argued that the knockout of AMPK usually occurred simultaneously with or even after tumorigenesis had been initiated. Altogether, the conflicting effects of AMPK on malignancies mentioned above provide a better understanding of the role of AMPK metabolic signalings in tumorigenesis and yield insights on innovative therapeutic strategies, including immunotherapy that targets the relevant metabolic networks in cancer.

## 4. Complementary Targeted Therapeutics—A Double Hit Effect on Ovarian Cancer

Different targeted therapies have been recently explored in ovarian cancer based on increasing knowledge of crucial biologic pathways driving tumor progression. Recently, breakthrough approvals have been made in many adjuvant drugs, including anti-VEGF antibodies [48,49]. For instance, mouse monoclonal anti-VEGF-A antibodies had been shown to significantly inhibit in vivo tumor growth and malignant ascites formation, leading to the development of Avastin [49]. Avastin is an FDA-approved monoclonal anti-vascular endothelial growth factor (VEGF)-A antibody targeting tumor angiogenesis that has been examined and commonly adopted for the regime of recurrent ovarian cancer for the past few years [48,49,50,51]. In ovarian cancers, VEGF is detectable by immunohistochemistry in primary lesions and malignant ascites and serum samples [49]. The expression of cancer-specific VEGF in malignant ascites that exceeded the 95th percentile of concentrations are frequently observed in benign diseases [49]. Multiple Phase II clinical trials have formerly evaluated Avastin as a doublet with other cytotoxic drugs that are regularly used for recurrent ovarian cancer. Overall response rate (ORR) varies between 24 and 50% in the studies of more than 20 patients. Progression-free survival (PFS) ranges from 6 to 8 months, and overall survival (OS) ranges from 13.8 to 33.2 months [49,51,52,53,54,55,56,57]. However, as reported in two large randomized clinical trials, i.e., Gynecologic Oncology Group (GOG) 218 [58] and ICON7 [59] trials, a modest improvement in PFS with no difference in OS was observed, indicating that there was no benefit of adding Avastin to existing chemotherapy in patients with newly diagnosed, incompletely resected, advanced ovarian carcinoma [58,59]. Thus, exploring alternative adjuvant therapeutic combinations is still an area of extensive and vital research.

Recent findings have demonstrated that activation of AMPK by different pharmaceutical activators such as Metformin and AICAR exhibits an inhibitory role in tumor growth along with angiogenesis [43,60,61,62,63]. Furthermore, natural AMPK activators such as bitter melon extract downregulate the expression of HIF1α and VEGF in hypoxic nasopharyngeal carcinoma cells [64,65]. At the same time, hop-derived flavonoid Xanthohumol exhibits strong angio-preventive activity, which is also mediated by the activation of AMPK in endothelial cells [43,66]. The most feasible way AMPK activation exerts a protective outcome to attenuate pathological angiogenesis is likely via mTOR inhibition [43]. Malignant cells are addicted to the dysregulated mTOR/HIF-1α/VEGF axis for their growth, pharmacological or natural AMPK activators, as reviewed above, are consistent with our previous studies that they can disrupt the associated signalings, leading to the inhibition of angiogenesis. Hence, activating AMPK signaling complementary to anti-VEGF therapy may generate a double hit effect to impede angiogenesis and tumorigenesis of ovarian cancer.

On the other hand, immunodepression is a common pathological phenomenon observed in the TME to promote tumor development and progression [67]. Immunotherapeutic strategies aim at reactivating the host antitumor immune system are therefore exploited as potential remedies for the management of a wide variety of malignancies [67,68,69]. Immune checkpoint blockade (ICB) is one of the promising denominator approaches among cancer immunotherapies [69]. The suppression of CD8^+^ T cell-mediated antitumor immune response can be significantly relieved by the immune checkpoint inhibitors (ICIs) such as anti-programmed cell death protein 1 (PD-1) and anti-PD-1 ligand 1 (PD-L1) [70,71]. In fact, an inverse correlation between the aberrant expression of PD-L1 on malignant cells and patient prognosis in ovarian cancer has been reported [72,73,74]. Nonetheless, responses to selective inhibition on PD-1/PD-L1 are limited to a fraction of ovarian cancer patients. Compared with the remarkable effects on other cancer types such as melanoma and NSCLC, the observed response in epithelial ovarian cancer trials has been modest at best. For instance, a Phase II clinical trial of Nivolumab, an FDA-approved anti-PD-1 antibody, in patients with platinum-resistant ovarian cancer displayed an overall response rate of only ~15% [18], suggesting that several challenges still exist in expanding the use of ICIs as effective intervention and exploration on potential therapeutics combination that can cooperate with the ICIs to advance the clinical response in ovarian cancer is urgently required.

Indeed, our latest studies identified that concomitant use of Glutaminase inhibitor 968 and anti-PD-L1 remarkably boosted the immune response against ovarian cancer [74]. Consistent with our findings, combined inhibition of PD-1 and activation of AMPK by Metformin, AICAR or BME had currently been proposed to double response rates in suppressing cell growth of different types of cancers and yielding a substantial increase in the number of tumor-infiltrating CD8^+^ T cells in mouse models [75,76,77,78], supporting the potential clinical use of AMPK activators when combined with anti-PD-1 ICIs. Additionally, the retrospective review disclosed better clinical outcomes (ORR, PFS, and OS) in patients with NSCLC who received concurrent Metformin and ICIs [79]. Further prospective studies assessing the long-term clinical benefits of Metformin when used in conjunction with PD-1 blockade, including a Phase Ib clinical trial (UMIN000028405) conducted at Okayama University on refractory/recurrent tumors of lung cancer, did demonstrate improved intratumoral T-cell function and tumor clearance [78,79]. Another Phase II clinical trial (NCT03048500) currently conducted in patients with unresectable advanced stages NSCLC was still in an active status, and the antitumor activity and combination efficacy should be evaluated (Table 1). Although PD-1 acts as an important negative feedback regulator of T cell effector functions, the upstream pathway implicated in the downregulation of PD-1 is still unclear. The discovery of the regulation between AMPK phosphorylation and repression of PD-1 is conceivably attributed to the AMPK downstream signalings such as the p38 MAPK/GSK3β axis [75] and the KEAP1/NRF2 cascade [76,80]. Of note, AMPK activation is thought not only to straightforwardly reduce tumor burden by inhibiting cancer cell growth but also to concurrently support the expansion and survival of tumor-infiltrating lymphocytes (TILs), especially CD8^+^ T cells within the TME, in part by the inhibition of glycolysis and promotion of oxidative phosphorylation [81,82,83,84]. For instance, AMPK activation by Metformin has been demonstrated to play an active role in metabolic reprogramming of TILs through increasing the population of CD8^+^ T cells, protecting them from apoptosis, and preventing their immune exhaustion characterized by a decreased secretion of IL-2, TNFα, and IFNγ [84,85,86]. Moreover, AMPK is reported to be a key regulator of T cell-mediated adaptive immunity by maintaining the consistent activity of TILs via activating oxidative metabolism upon nutrient deprivation [87]. Similarly, mice with a T cell-specific ablation of TSC2, a negative regulator of mTORC1, resulted in the generation of highly glycolytic CD8^+^ T cells that were incapable of transitioning into a memory state [88]. In fact, mounting evidence has indicated that AMPK/mTOR signaling is responsible for mediating memory CD8^+^ T cells differentiation. It has been revealed that AMPKα1-deficient T cells exhibit a defect in the generation of memory CD8^+^ T cells [86,89]. Consistently, rapamycin-mediated mTOR repression enhanced memory CD8^+^ T cell responses in virus-infected murine and non-human primate models [90]. Accordingly, AMPK-directed therapeutics should carefully consider both the direct effects on cancer cell death and the immune cell phenotype effects within the TME (Figure 2). Collectively, these outcomes have paved the way for several combination studies with diverse AMPK activators and systemic ICIs and hopefully can improve the efficacy of immunotherapy in ovarian cancer.

In spite of the many ongoing studies about the combo therapy for cancer treatment, the effectiveness and translational use of the dual-targeted remedy in the clinic are still limited. The selection of cancer cells always occurs when they acquire adaptation to the drug treatment by modifying drug metabolism and altering drug-target interactions [91,92]. Furthermore, the survival of tumor cells is supported by multiple signal transductions that the repression of one of them may promptly trigger the activation of the compensatory signaling pathways to reestablish the induction of downstream signals regardless of the hampered original oncogenes, making optimization of dosages and sequences of the drug combination against resistance mechanisms more discouraging [91,93,94,95]. Intra-tumor heterogeneity, especially the presence of cancer stem cells ubiquitously observed in ovarian cancer, adds another dimension of complexity to the success of the targeted therapy, as it is pretty challenging to target all the driver mutations from numerous sub-clones of ovarian cancer cells [92,95,96]. Recently, it has been disclosed that various sub-clones of metastatic ovarian cancer cells exhibited increased aggressiveness in the ascites microenvironment via reprogramming fatty acid metabolism [23]. Since lipid metabolism can remodel the immune function and sensitivity to ferroptosis [97,98], additional strategies are required to determine whether combined AMPK activation with targeting lipid metabolism signalings and ferroptosis is an effective anti-cancer therapy.

## 5. Targeting Fatty Acid Metabolism for Cancer Immunotherapy

Most malignant cells, including ovarian cancer cells, exhibit an aberrantly upregulated lipid metabolism, which allows them to biosynthesize and desaturate fatty acids to support cancer proliferation [2,99,100,101]. Our previous study likewise revealed that lipid-enriched ascites enforced ovarian cancer cells to undergo metabolic reprogramming and utilized fatty acids as a significant energy source for tumor aggression and development [2,23], suggesting particular subsets of ovarian cancer cells might be sensitive toward approaches targeting lipid metabolism. While inhibiting fatty acid metabolism in cancer cells may induce individual consequences on different immune cell subsets, lipid accumulation usually leads to immunosuppressive effects. Among bone marrow-derived myeloid cells, the presence of the long-chain unsaturated fatty acid such as oleate promotes immunosuppressive function, shifting the polarization of tumor-associated macrophages (TAMs) into a protumorigenic M2-like phenotype [102,103]. In the ovarian cancer ascites, unsaturated fatty acids, including linoleic acid (LA) and arachidonic acid (AA) can be transformed into prostaglandin E2 (PGE2) by COX-2, which can induce the secretion of CXCL12 and the expression of CXCR4 to promote the accumulation of myeloid-derived suppressor cells (MDSCs) and inhibition of T cell functions [22,25]. Accumulation of fatty acids thus has significant influences among the immune cell populations within the TME, and strategies aimed to ameliorate lipid abundance through targeting fatty acid synthesis (FAS) or fatty acid oxidation (FAO) may benefit outcomes. For example, FAO is required for the development of memory CD8^+^ T cells and the differentiation of regulatory T cells, so manipulation of lipid capacity via fatty acid oxidation probably has the potential to select the deposit of specific T cell populations [82,104,105,106]. Furthermore, studies have reported that PPAR-induced β oxidation in TILs raises the number of active effector CD8^+^ T cells and subsequently facilitates anti-PD-1 therapy [25,107]. Actually, unsaturated fatty acids in the malignant ascites can also modulate the behavior of macrophages through PPARs that may indirectly affect the function of T cells [25]. On the other hand, pharmaceutical and genetic suppression of fatty acid synthase (FASN) has been shown not only to protect T cells from apoptosis within the TME triggered by repeated T cell receptor (TCR) activation but also to boost T cell immunity along with its antitumor efficacy on tumor cells [82,108]. Coincidentally, TCGA analysis unveiled that amplification alteration in the lipogenic enzymes participating in the fatty acid biosynthesis pathway such as ACACA, ACACB, FASN, and MCAT among serous subtypes of ovarian cancer was the most abundant genetic alteration detected (Figure 3A), suggesting that high lipogenesis supports oncogenic properties of ovarian cancer. More importantly, the TIMER2.0 database algorithm used to explore the correlation between the expression of the abovementioned lipogenic enzymes and the infiltration of immune cells revealed that these FAS-related genes were negatively associated with the infiltration of CD8^+^ T cells among ovarian cancer patients (Figure 3B), inferring an inverse connection of lipogenesis with the availability of effector T cells for effective antitumor efficiency in ovarian cancer. In line with these findings, our previous investigation disclosed that the combined treatment of low toxic AMPK activators with TAK1 and FASN inhibitors synergistically impairs oncogenic augmentation of ovarian cancer [23]. Collectively, alteration of fatty acid metabolism to relieve lipid abundance does offer intriguing opportunities to modulate different types of immune cells within the TME and subsequently remodel their immune function (Figure 4).

## 6. Ferroptosis Initiating Therapies (FITs) as the Achilles Heel in Cancer Treatment

Ferroptosis is a novel means of regulating cell death caused by an accumulation of lipid-based oxidation products in an iron-dependent manner [109,110,111]. Cancer cells escaping from other regulated forms of cell death such as apoptosis or autophagy during tumor development still maintain sensitivity to ferroptosis, implying that inducing ferroptosis could be a therapeutic strategy for anti-cancer treatment [110,112]. It has been hypothesized that metabolic reprogramming leads to acquiring ferroptosis sensitivity as part of an escape strategy against other therapies [113,114], proposing a possible application of FITs in the management of persister cancer cells and MRD [113,115]. Given that the high lipid metabolic activities support oncogenic properties of metastatic ovarian cancer cells, it is conceivable that the devastating accumulation of iron-dependent lipid peroxidation products is considered to be the main executioner triggering ferroptotic cell death if the antioxidant defense systems are being overwhelmed. In accordance with this concept, our latest study also verified that the augmented cell growth, membrane fluidity, cancer stem cell formation, and EMT of ascites-derived ovarian cancer cells were attributed to the overexpression of two fatty acid desaturases (FADs), SCD1 and FADS2 [116]. Combined inhibition of SCD1/FADS2 in ascites-derived ovarian cancer cells abrogated the GSH-GPX4 system, disrupting the cellular/mitochondrial redox equilibrium and, eventually, ferroptotic cell death. Significantly, harnessing commercially available inhibitors to selectively suppress SCD1/FADS2 activities enhanced the synergistic anti-cancer effect of Cisplatin to promote tumor clearance [116]. These findings not only provide novel and alternative therapeutic regimens for combating chemoresistance and peritoneal metastasis of ovarian cancer but also indicate FITs are of great biological significance and clinical relevance.

While substantial research has been focused on the effect of ferroptotic damage in malignant cells for years, the immunogenic features of ferroptosis between tumor niche and cancer immunity have remained rarely explored. Thus far, it is conceivable that ferroptotic cells can release distinct “find me” signals such as lipid mediators to attract antigen-presenting cells (APCs) and other immune cells to the niche with ferroptotically dying cells, resulting in the engulfment by macrophages in a mechanism different from the phagocytic clearance of apoptotic cells [114,117,118]. On the other hand, it has been examined that the release of ATP and high mobility group box 1 (HMGB1) from the early ferroptosis tumor cells can be recognized as damage-associated molecular patterns (DAMPs) by certain immune cells to foster antitumor immunity [119]. DAMPs behaved as immune modulators that mainly attract and activate APCs or dendritic cells (DCs) for the uptake of tumor-associated antigens (TAAs). After processing by MHC class I-restricted cross-presentation of these TAAs, priming of T cells takes place, which leads to clonal expansion of cancer-specific cytotoxic T lymphocytes [120], supporting the notion that ferroptosis is implicated in the recruiting and stimulating functions of immune cells upon immune responses and immunotherapies (Figure 4).

Given that T cells are critical mediators of antitumor immunity, ferroptosis has recently been demonstrated to participate in ICIs (e.g., anti-PD-1 or anti-CTLA4)-mediated antitumor immune responses driven by cytotoxic T cells. INF-γ that is secreted from the activated cytotoxic CD8^+^ T cells upon ICIs blockade enhances ferroptotic effect on tumor cells by downregulating the expression of cysteine/glutamate transporter (system X_C_^−^) subunits, SLC3A2 and SLC7A11, leading to reduced cystine uptake and subsequently enhanced lipid peroxidation and ferroptosis of the cancer cells [111,121,122] (Figure 4).

With the promising impact of cancer immunity upon ferroptosis induction, it is expected to be a beneficial anti-cancer modality to improve current cancer treatment. Nonetheless, ferroptosis in immune cells themselves may compromise the immune responses, and there is considerable controversy as to whether ferroptosis is a two-edged sword. It has been aware that GPX4 activity is not only important for the survival of tumor cells but also plays an essential role in the development of lymphocytes [122,123]. Further works on T cells have revealed that deficiency of GPX4 in antigen-specific CD4^+^ and CD8^+^ T cells are unable to expand and experience ferroptotic cell death induction with an accumulation of lipid peroxides, thereby precluding their immune response to protect the host from infections [114,123]. Another proposed strategy for boosting the antitumor activity by ferroptosis is iron modulation [124]. However, iron addiction is one of the distinct characteristics of cancer cells that they evolve multiple mechanisms to concomitantly increase iron uptake and decrease iron export so as to ensure the iron supply for cell proliferation and aggression [125,126]. In addition, cancer cells with dysregulated iron metabolism can reconfigure immune cells within the TME, for example, by revising the polarization of TAMs to an anti-inflammatory phenotype associated with the augmented release of iron, leading to enhanced tumorigenesis and immunosuppression [122,127]. Taking into account the iron-overdosed properties around the cancerous neoplasm, ferroptosis induction promoted by delivering iron into the TME as an adjuvant may escalate cancer progression and immune evasion. Nevertheless, barriers to ferroptosis induction on the potent antitumor immunity are not limited to these. Ferroptosis in cancer cells has been reported to be associated with strengthened secretion of immunosuppressive agents such as prostaglandin E2 (PGE2) [122,128], which stimulates cancer cell growth and represses cytotoxic T cell activity, inferring that further understanding of the complicated crosstalk among cancer and immune cells as well as the dual role of ferroptosis in tumor immunity is required to provide new insight on targeting ferroptosis in cancer immunotherapy.

As ferroptosis is a complex process governed by multiple layers of metabolic signaling pathways, targeting ferroptosis in an immunotherapeutic approach and metabolic rewiring is considered a potential strategy. Recent studies have revealed a critical role of the fatty acid transporter CD36 in promoting ferroptotic cell death in T cells. The results showed that overexpression of CD36 in CD8^+^ TILs advanced the uptake of oxidized lipids and promoted lipid peroxidation, inducing ferroptosis in CD8^+^ T cells and lowering the secretion of INF-γ into the TME [129,130]. These findings not only uncover the feasibility of targeting lipid metabolism to overcome the immunosuppression from CD8^+^ T cell ferroptosis, but also underscore the therapeutic potential of modulating fatty acid uptake via blocking CD36 to boost anti-cancer immunity. On the other hand, AMPK is a cellular energy sensor that plays an important role in reprogramming cancer cell metabolism in various human cancers [23,131]. Recent results showed that system X_C_^−^ inhibitors Erastin and Sulfasalazine induced the AMPK-mediated phosphorylation of Beclin 1 to form Beclin 1-SLC7A11 complex and subsequent ferroptosis initiation [122,132,133]. In contrast, metabolic proteins suppressing ferroptosis, such as BCAT2, were found to be inhibited by the AMPK/SREBP1 signaling upon treatment of ferroptosis inducers (Erastin, Sorafenib, and Sulfasalazine) in HCC cells [122,134], suggesting that induction of ferroptosis together with attention to cancer cell metabolism via modulation of AMPK activity may be an emerging anti-cancer strategy to combat tumors. Aforementioned, we have demonstrated that co-inhibition of SCD1/FADS2 enhanced not only ferroptotic cell death in metastatic ovarian cancer cells with high demand on lipid metabolism but also sensitizing ovarian cancer cells to Cisplatin-induced cell cytotoxicity [116]. Similar synergizing effects of ferroptosis induction in combination with common therapeutic agents have gradually been reported, for example, pre-treatment of Erastin followed by Cisplatin-induced cell death in a variety of cancer cell lines [135]. Furthermore, Altretamine, an inhibitor of GPX4 lipid repair activity, has already been exploited in ovarian cancer treatment that exhibited ferroptosis initiation and showed well tolerance and associated with prolonged PFS and OS in Phase II clinical study [122,136,137]. Therefore, it should be optimistic about the upcoming management of ovarian cancer that promising translational anti-cancer strategies would be developed, and novel combination remedies based on FITs would be helpful to improve patient outcomes.

## 7. Expanded Perspectives

During tumor development, myriad oncogenic pathways such as DVL3/Wnt/β-catenin, AKT/ERK/FOXM1, and mTOR/p70S6K signals etc. are constitutively activated in cancer cells. AMPK, which is the pivotal signaling hub responsible for regulating these processes within the cells, is becoming an important target for cancer therapy. In fact, the therapeutic efficacy of AMPK activation has long been recognized in many types of malignancies [98,138,139,140]. The effectual outcomes of AMPK induction in impeding tumor progression thus open a new door to advocating different combination strategies, including those duple-targeted remedies mentioned above, in the treatment of chemoresistance and carcinomatosis of ovarian cancer. However, cancer cells are believed to have a great deal of cunning in that they can versatilely adjust the molecular and cellular mechanisms for survival under the pressure of drug perturbation, leading to the evolvement and repopulation of more aggressive or metastatic phenotypes which are no longer sensitive to the treatment. Our recent finding identified that BCL2A1, an inducible BCL2 member, not only protects cancer cells against cellular stress-mediated intrinsic (mitochondrial) apoptosis but also promotes tumor growth and metastatic progression in ovarian cancer peritoneal metastases [141]. In fact, the diverse heterogenous sub-clones of malignant cells, together with the surrounding cancer-associated fibroblasts (CAFs) and immune cells, form an ecosystem, cooperating and opposing each other for nutrients and spaces from the harsh TME [91]. Consequently, it is conceivable that narrowing the population dynamics of tumor cells via tuning the metabolic processes and TME may be a powerful strategy to additionally ameliorate the targeted remedies in eliminating ovarian cancer [142].

Metabolic reprogramming has been recognized as a hallmark of cancer due to its significance for metastatic cancer progression and MRD [143]. Our previous publication indeed revealed that the metastatic ovarian cancer cells underwent metabolic reprogramming to utilize lipid metabolism for tumor progression in the fatty acid-enriched microenvironment of the peritoneal cavity through the TAK1/NF-κB signaling [23], evincing particular subsets of ovarian cancer cells might be susceptible toward approaches targeting lipid metabolism. The results from the study had not only confirmed that TAK1/NF-κB signaling, in line with mTOR, was negatively regulated by AMPK in ovarian cancer cells, but also provided an engaging example that AMPK activator could be potently exploited with a combined cocktail of TAK1 inhibitor 5Z-O and lipogenesis modulator Orlistat to exert synergistic anti-cancer effects in both in vitro 3D spheroids culture of ovarian cancer cells and in vivo metastatic dissemination of ovarian cancer. Although the clinical benefit of the AMPK activation-mediated targeted therapy with a combined cocktail of lipid metabolic modulators is still reticently reported, the innovative idea proposed herein may recommend the feasibility of this alternative therapeutic intervention to impede ovarian cancer peritoneal metastases.

Given the intimate association between fatty acid metabolism and ferroptosis, it is believed that signaling cascades that intermediate fatty acid metabolic processes may perhaps serve a vital role in impinging on the tolerance of cancer cells toward lipid peroxidation and ferroptosis induction. Ferroptosis induction may leave us with distinctive potential to preferably eradicate certain sub-clones of cancer cells [114], especially those in a high-mesenchymal cell state [115,144] and those on the run of drug treatment [144,145]. A deeper understanding of the metabolic underpinnings that orchestrate ferroptosis is mandatory to harness its full pharmacological potential to tackle ovarian cancer [144].

## 8. Conclusions

In summary, numerous studies nowadays propose alternative therapeutics either alone or in combination with conventional therapies for better management of different malignancies. Many adjuvant drugs, such as anti-VEGF/PD-1 antibodies, are already in clinical trials or applications. With the strategic combination of metabolic modulators, especially pharmaceutical/natural AMPK activators with low adverse effects and cost-effectiveness, this will be concomitantly beneficial to suppress the oncogenic pathways/metabolisms and boost the body’s immune defense system to prevent ovarian cancer risk. Hence, it is worth inputting more endeavors onto further mechanistic evaluations to validate these recommended hypotheses, and hopefully, discussions herein would shed light on the application of AMPK activators, VEGF/PD-1 blockades, and FITs in the treatment of human ovarian cancer.

## Figures and Tables

**Figure 1 ijms-23-06857-f001:**
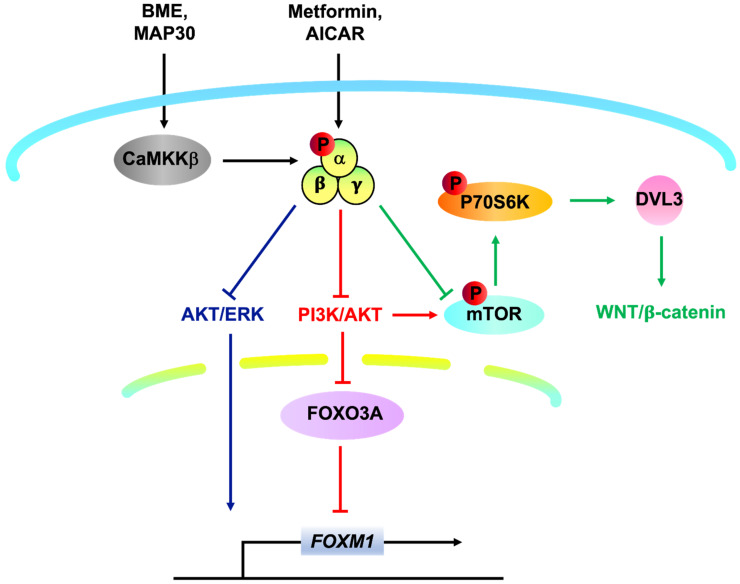
The significance of targeting AMPK in human ovarian cancer. AMPK is a universal heterotrimeric serine/threonine protein kinase composed of a catalytic subunit α and two regulatory subunits β and γ. Each subunit has different isoforms such as α1, α2, β1, β2, γ1, γ2, and γ3 encoded by distinct genes, which enables the yielding of in total twelve possible heterotrimeric combinations. AMPK activation in response to pharmaceutical (Metformin, AICAR etc.) and natural (BME, MAP30 etc.) AMPK activators have been shown to exert ant-tumor effects in different types of human carcinoma, including ovarian cancer, by modulating a variety of oncogenic pathways such as AKT/ERK/FOXM1 (Blue), AKT/FOXO3a/FOXM1 (Red), mTOR/p70S6K/DVL3/WNT/β-catenin (Green) signals.

**Figure 2 ijms-23-06857-f002:**
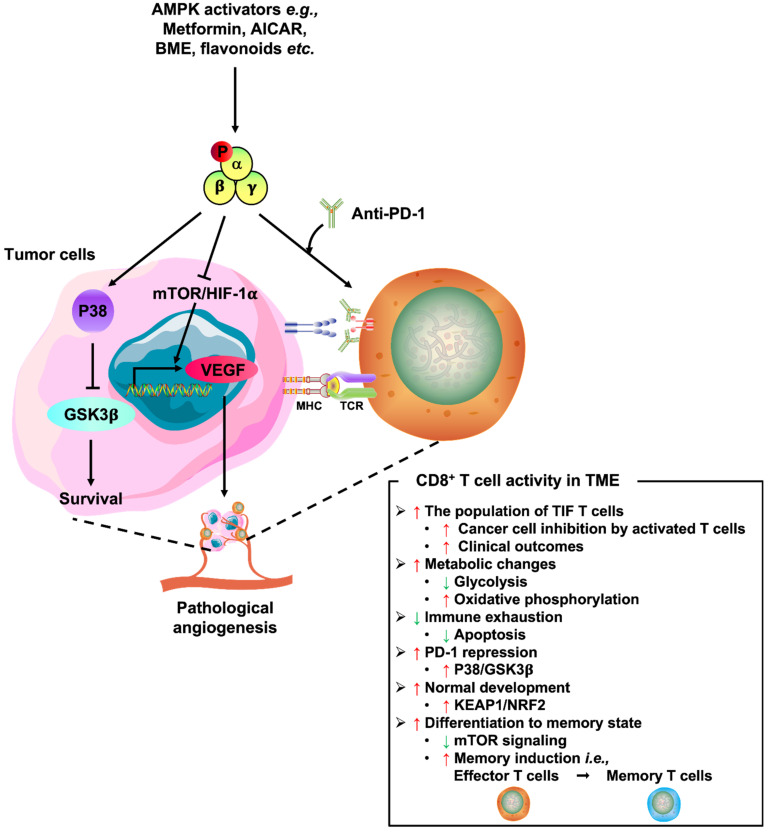
Several beneficial effects of AMPK activators on antitumor immunity. AMPK activation not only inhibits cell growth and survival in cancer cells per se, but it also stimulates TILs CD8^+^T cell activity within the tumor microenvironment (TME) by enhancing the CD8^+^ T cell residents, preventing their immune exhaustion, maintaining their consistent activity via activating oxidative metabolism and facilitating their transition into the memory state.

**Figure 3 ijms-23-06857-f003:**
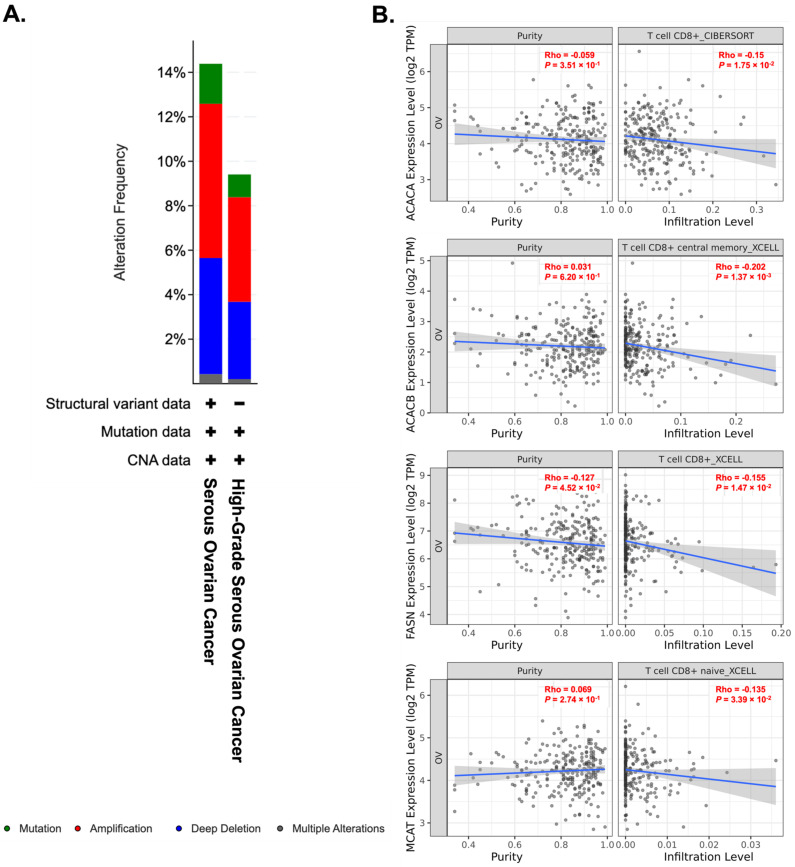
Targeting lipogenesis may exert beneficial effects in alleviating the impaired immune cell infiltration. (**A**) By means of the cBioPortal online tool, the genetic alterations of different lipogenic enzymes such as ACACA, ACACB, FASN, and MCAT in ovarian cancer patients were analyzed and summarized in the bar chart. Amplification alteration was the most frequently detected genomic alteration among serous subtypes of ovarian cancer. (**B**) Inverse correlations between differential expression of different lipogenic enzymes and abundance of CD8^+^ T cell infiltration in ovarian cancer by TIMER algorithm using datasets from Cibersort and XCell.

**Figure 4 ijms-23-06857-f004:**
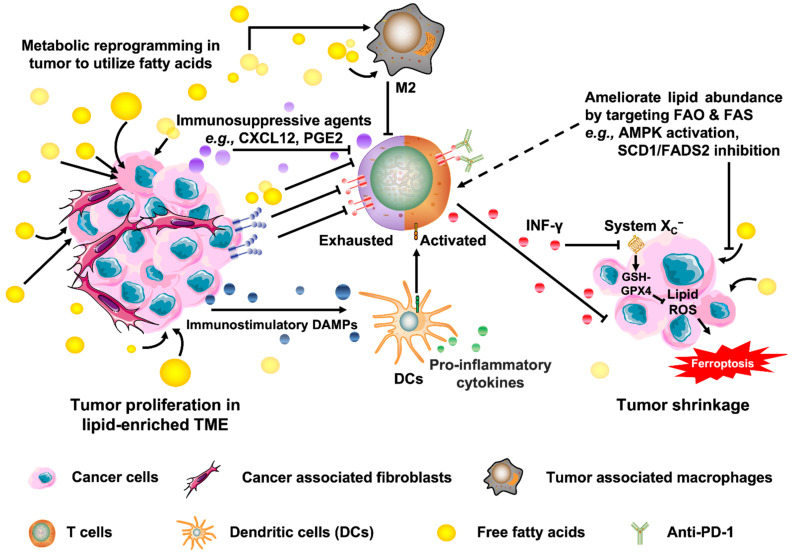
A schematic diagram to illustrate the metabolic alternations and immunogenic features of ferroptotic cancer cells. On the one hand, targeting fatty acid metabolism for cancer immunotherapy as the proposed molecular mechanism to down-regulate the accumulation of fatty acids in cancer cells and TME and subsequently inhibit tumor growth. On the other hand, ferroptosis has been proposed to play a significant role in mediating various functions during immune responses. Ferroptotic cancer cells release DAMPs such as ATP and HMGB1 that act as immune modulators stimulating the maturation of antigen-loaded DCs. The activated DCs subsequently release pro-inflammatory cytokines and present TAA to activate CD8^+^ T cells, which release IFN-γ and downregulate system X _C_^−^ and eventually induce ferroptosis.

**Table 1 ijms-23-06857-t001:** Information about the clinical trials on the combined treatment of Metformin and ICIs from NIH ClinicalTrials.gov. PD-1 inhibitors Nivolumab, Pembrolizumab, and Cemiplimab are on the current list of the FDA-approved agents. However, only the first two of them are being explored in concurrent treatment with the AMPK activator Metformin.

Clinical Trials	No. of Patient	Disease	Intervention/ Treatment	Phase	Recruitment Status	Sponsor
Nivolumab and Metformin Hydrochloride in Treating Patients with Stage III-IV Non-small Cell Lung Cancer That Cannot Be Removed by Surgery (NCT03048500)	N = 17	NSCLC (Stage III, IIIA, IIIB & IV)	Metformin & Nivolumab	Phase II	Active, not recruiting	Northwestern University
Nivolumab and Metformin in Patients with Treatment Refractory MSS Colorectal Cancer (NCT03800602)	N = 24	Colorectal Cancer (Stage IVA, IVB & IVC)	Metformin & Nivolumab	Phase II	Active, not recruiting	Emory University
Anti-PD-1 mAb Plus Metabolic Modulator in Solid Tumor Malignancies (NCT04114136)	N = 108	Melanoma, NSCLC, HCC, Urothelial Cancer, Gastric Adenocarcinoma, HNSCC, Esophageal Adenocarcinoma & MSI-High solid tumors	Metformin & Nivolumab or Pembrolizumab (dependent upon approved indication)	Phase II	Recruiting	Dan Zandberg, University of Pittsburgh
Combining Pembrolizumab and Metformin in Metastatic Head and Neck Cancer Patients (NCT04414540)	N = 20	HNSCC	Metformin & Pembrolizumab	Phase II	Recruiting	Trisha Wise-Draper, University of Cincinnati
A Trial of Pembrolizumab and Metformin Versus Pembrolizumab Alone in Advanced Melanoma (NCT03311308)	N = 30	Advanced Melanoma	Metformin & Pembrolizumab	Phase I	Recruiting	Yana Najjar, University of Pittsburgh

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
