# Peer review of "Orchestrated Action of AMPK Activation and Combined VEGF/PD-1 Blockade with Lipid Metabolic Tunning as Multi-Target Therapeutics against Ovarian Cancers"

_ijms, 2022, doi:10.3390/ijms23126857_

Round 1
Reviewer 1 Report
This mini-review focused on the combination of AMPK activators coupled with VEGF/PD-1 blockade as a dual-targeted therapy against the most lethal gynecological malignancy ovarian cancer. It is a well-written manuscript in general, but some minor issues require authors' attention.
- to cite more recent papers as some are relative outdated,
- to include outcomes of recent clinical trials as well as possible mode-of-actions,
- to discuss the limitation of such combinations as a dual-targeted therapy, and
- to provide a brief comparison among these alternative combo therapies.
Author Response
- To cite more recent papers as some are relative outdated.
Response: More recent references have been cited in the revised paragraphs of the manuscript.
- To include outcomes of recent clinical trials as well as possible mode-of-actions.
Response: Most of the clinical trials about the dual-targeted therapy written in the manuscript are still in progress, and so the outcomes are reticently made public.
- To discuss the limitation of such combinations as a dual-targeted therapy.
Response: A new paragraph about the limitation has been added following the original paragraph about dual-targeted therapy.
- To provide a brief comparison among these alternative combo therapies.
Response: A new section, Section 7, has been added to discuss and extend the idea of these alternative combo therapies.
Reviewer 2 Report
I enjoyed reading the review article by Yung et al. on Ovarian cancers related therapies, focused on dual-targeted therapies combined with VEGF/PD-1 blockade. The review is comprehensive, detailed, and adequately cites recent literature. The main concerns I had was about the organization of the manuscript, which I am listing below. These are addressable concerns.
1. The authors indicated several approaches ranging from AMPK activation, to ferroptosis, to fatty acid metabolism targeting. These are all interesting approaches, but first they are not reflected in the title (which is focused primarily on AMPK activation), and seemed a little disorganized. Particularly jarring was grouping FAM and FIT within "Future Prospects". There is no intellectual buildup in the rest of the review about these, and suddenly the future prospects presents entirely new lines of mechanisms (instead of, for example, addressing particular challenges identified in AMPK activation).
I would suggest that the whole review is organized section wise differently. If FAM and FIT must be included, then it could be of the same heirarchical arrangement as AMPK activation.
2. Please include a section towards the end wherein mechanisms related to AMPK, FAM, and FIT are integrated together to justify their inclusion in the same review article. Otherwise it seems that the review reflects whatever the authors were interested in. This could be achieved, as the authors have written about AMPK activation and FIT together in Section 5.2
3. The legends of the figures are very inadequate, and seemed to comprise merely of the titles of the figures.
4. If the authors want to build AMPK activation in greater detail than other "dual mechanisms", then they should make attempts to arrange the article with increased structure. Section 4, for example, extends to 2 full pages in small font, and it seemed a lot under the sky is covered there without much synthesis. It seems like a literature review, which was a major concern all through the manuscript. A good review should extend significantly more than a literature review, which even a charitable viewing prohibited from crediting this manuscript.
I welcome the review, the details and extensive references, but the review really needs to be better organized. It is too flat a structure, with little synthesis of ideas beyond a compendium of AMPK/FIT/VEGF targeting therapies and what they did. Do they interact together, is one likely to work better than others, is there some idea about segregation of populations where one therapy might work better than the other. What are the challenges each dual therapy faces, and if there are genetic bases for these challenges (or microenvironmental) etc etc. A lot of such questions are the ones which must be thought about carefully to integrate, and synthesize a review so that the whole is greater than the parts.
Author Response
- The authors indicated several approaches ranging from AMPK activation, to ferroptosis, to fatty acid metabolism targeting. These are all interesting approaches, but first they are not reflected in the title (which is focused primarily on AMPK activation), and seemed a little disorganized. Particularly jarring was grouping FAM and FIT within "Future Prospects". There is no intellectual buildup in the rest of the review about these, and suddenly the future prospects presents entirely new lines of mechanisms (instead of, for example, addressing particular challenges identified in AMPK activation).
I would suggest that the whole review be organized section-wise differently. If FAM and FIT must be included, then it could be of the same hierarchical arrangement as AMPK activation.
Response: According to the suggestions from the reviewer, the organization of the manuscript has been wise differently revised, with FAM and FIT being separated from individual sections, and the title has been amended so that it reflects the content of the writing.
- Please include a section towards the end wherein mechanisms related to AMPK, FAM, and FIT are integrated together to justify their inclusion in the same review article. Otherwise, the review reflects whatever the authors were interested in. This could be achieved, as the authors have written about AMPK activation and FIT together in Section 5.2.
Response: A new section, Section 7, has been added to justify their inclusion in the same review article.
- The legends of the figures are very inadequate and seemed to comprise merely of the titles of the figures.
Response: More detailed figure legends are added.
- If the authors want to build AMPK activation in greater detail than other "dual mechanisms", then they should make attempts to arrange the article with increased structure. Section 4, for example, extends to 2 full pages in small font, and a lot under the sky is covered there without much synthesis. It seems like a literature review, which was a major concern all through the manuscript. A good review should extend significantly more than a literature review, which even a charitable viewing prohibited from crediting this manuscript. I welcome the review, the details and extensive references, but the review really needs to be better organized. It is too flat a structure, with little synthesis of ideas beyond a compendium of AMPK/FIT/VEGF targeting therapies and what they did. Do they interact together, is one likely to work better than others, is there some idea about segregation of populations where one therapy might work better than the other. What are the challenges each dual therapy faces, and if there are genetic bases for these challenges (or microenvironmental) etc etc. A lot of such questions are the ones which must be thought about carefully to integrate and synthesize a review so that the whole is greater than the parts.
Response: A new paragraph about the limitation has been added in Section 4 following the original paragraph about the dual-targeted therapy to extend to 2 full pages. In addition, Section 7 has been added to discuss and extend the idea of these alternative therapies.
Round 2
Reviewer 2 Report
The authors have addressed the concerns raised.